# Interaction Analysis between the *Arabidopsis* Transcription Repressor VAL1 and Transcription Coregulators SIN3-LIKEs (SNLs)

**DOI:** 10.3390/ijms23136987

**Published:** 2022-06-23

**Authors:** Chuanyou Chen, Xia Gong, Yan Li, Haitao Li, Haitao Zhang, Li Liu, Dacheng Liang, Wenya Yuan

**Affiliations:** 1State Key Laboratory of Biocatalysis and Enzyme Engineering, School of Life Sciences, Hubei University, Wuhan 430062, China; ccy@stu.hubu.edu.cn (C.C.); gongxia15927540162@163.com (X.G.); 20210035@hubu.edu.cn (Y.L.); lht@hubu.edu.cn (H.L.); zht@hubu.edu.cn (H.Z.); liuli2020@hubu.edu.cn (L.L.); 2Hubei Collaborative Innovation Center for Grain Industry, School of Agriculture, Yangtze University, Jingzhou 434023, China; dachengliang@yangtzeu.edu.cn

**Keywords:** VAL1, SNLs, CW domain, PAH domain, protein–protein interaction

## Abstract

*VIVIPAROUS1/ABSCISIC ACID INSENSITIVE3-LIKE1* (*VAL1*) encodes a DNA-binding B3 domain protein and plays essential roles in seed maturation and flowering transition by repressing genes through epigenetic silencing in *Arabidopsis*. *SWI-INDEPENDENT3* (*SIN3*)-*LIKEs* (*SNLs*), which encode scaffold proteins for the assembly of histone deacetylase complexes and have six SIN3 homologues (*SNL1**–SNL6*) in *Arabidopsis thaliana*, directly repress gene expression to regulate seed maturation and flowering transition. However, it remains unclear whether VAL1 and SNLs work together in repressing the expression of related genes. In this study, yeast two-hybrid and firefly luciferase complementation imaging assays revealed that VAL1 interacts with SNLs, which can be attributed to its own zinc-finger CW (conserved Cys (C) and Trp (W) residues) domain and the PAH (Paired Amphipathic Helices) domains of SNLs. Furthermore, pull-down experiments confirmed that the CW domain of VAL1 interacts with both intact protein and the PAH domains of SNLs proteins, and the co-immunoprecipitation assays also confirmed the interaction between VAL1 and SNLs. In addition, quantitative real-time PCR (qRT-PCR) analysis showed that *VAL1* and *SNLs* were expressed in seedlings, and transient expression assays showed that VAL1 and SNLs were localized in the nucleus. Considered together, these results reveal that VAL1 physically interacts with SNLs both in vitro and in vivo, and suggest that VAL1 and SNLs may work together to repress the expression of genes related to seed maturation and flowering transition in *Arabidopsis*.

## 1. Introduction

Eukaryotic DNA is compacted into chromatin within the nucleus. Chromatin is a dynamic structure being constantly “open” and “closed” by transcription factors to regulate gene expression through epigenetic mechanisms, which is generally initiated in response to intrinsic and/or external signals and coordinated networking between transcriptional regulators, coregulators and chromatin modification factors. One mechanism for a cell to regulate the dynamics of chromatin is to alter the acetylation levels of histones, which are the core components of nucleosomes. It has been well documented that histone acetylation is correlated with gene expression, while histone deacetylation is associated with gene repression.

VIVIPAROUS1/ABSCISIC ACID INSENSITIVE3-LIKE1 (VAL1) and its homolog VAL2 are transcription factors that can bind to the RY *cis*-element (CATGCA/TGCATG) of the *FLOWERING LOCUS C* (*FLC*) and *FLOWERING LOCUS T* (*FT*) locus via its DNA-binding B3 domain in *Arabidopsis* [1,2,3,4,5,6]. VAL1 and VAL2 were initially identified to play an essential role in the transition from embryonic to vegetative growth by repressing the genes related to seed maturation [7,8,9,10,11,12,13], and recently they were found to be involved in flowering transition by repressing the flowering genes *FLC* and *FT* in *Arabidopsis* [2,3,6]. VAL1/2 function as transcription repressors of genes through different epigenetic mechanisms. VAL1 interacts with AtBMI1A and AtBMI1B, the components of plant Polycomb repressive complex 1 (PRC1), to recruit PRC1 to initiate H2A lysine 121 ubiquitination (H2A121ub) in seed maturation genes [9]. VAL1/2 interact with LIKE HETEROCHROMATIN PROTEIN 1 (LHP1), a putative member of plant Polycomb repressive complex 2 (PRC2), to recruit PRC2 to facilitate H3K27me3 accumulation at *FLC* [2,3]. VAL2 interacts with HISTONE DEACETYLASE19 (HDA19), a core component of the deacetylation complex, to mediate histone deacetylation at the target loci [13]. In addition to the plant-specific B3 DNA-binding domain, VAL1/2 contain three other domains, including the plant homeodomain (PHD), the zinc-finger CW domain (CW) and the ethylene-responsive element binding factor-associated amphiphilic repression domain (EAR) [3,7]. The PHD domain and CW domain are potential histone-binding domains and mediate the protein–protein interactions. The PHD domain of VAL1 can bind to di- and tri-methylated H3K27 [3] and the CW domain of VAL2 is responsible for the interaction between VAL2 and HDA19 [13]. The EAR domain is an active repression domain, which also mediates protein–protein interactions [14]. 

The general transcription coregulator SIN3 is a scaffold protein of a multisubunit protein complex typically associated with the histone deacetylase (HDAC) and has been shown to alter gene expression in many eukaryotes [15]. In *Arabidopsis,* there are six homologs of *SIN3* designated as *SIN3-LIKEs* (including *SNL1* to *SNL6*), which work redundantly in different developmental stages [16]. SNL1 and SNL2 redundantly regulate key genes involved in the ethylene and abscisic acid (ABA) pathways by decreasing their histone acetylation levels to regulate primary seed dormancy [17], and affect the speed of seed germination through SNL histone deacetylase-binding factor-mediated regulation of *AUXIN RESISTANT 1* (*AUX1*) [18]. *SIN3-LIKE* genes induce flowering under long-day conditions but inhibit floral transition under short-day conditions by repressing the expression of an *FT* activator and three *FT* repressors [19]. SIN3 is thought to mediate multiple protein–protein interactions via its paired amphipathic helix (PAH) domains and an HDAC interaction domain (HID). SIN3 has no demonstrated intrinsic DNA-binding activity, but is recruited to the target DNA through interactions with site-specific DNA-binding factors to achieve its gene-specific regulatory activity [16]; however, it remains largely unknown what DNA-binding factors are responsible for the recruiting of SIN3 to the corresponding target loci.

Our previous studies have revealed that both VALs and SNLs are involved in flowering transition by repressing the expression of related genes through histone deacetylation [3,19]. Therefore, we speculate that they may work together when repressing certain genes. In this study, yeast two-hybrid (Y2H) assay, pull-down assay, firefly luciferase complementation imaging (LCI) assay and co-immunoprecipitation (Co-IP) assay were performed to validate that VAL1 directly interacts with SNLs (SNL1–SNL6), which could be ascribed to the CW domain of VAL1 and the PAH domains of SNLs. In addition, quantitative real-time PCR (qRT-PCR) analysis showed that *VAL1* and *SNLs* were co-expressed in *Arabidopsis* seedlings, and transient expression analysis in *Arabidopsis* protoplasts showed that VAL1 and SNLs were co-localized in the nucleus. Based on the results, it can be speculated that the SIN3-containing HDAC complex may be recruited to the target genes through the DNA-binding transcription factor VAL1 to be involved in seed maturation and flower transition.

## 2. Results

### 2.1. Y2H Assay Demonstrated That VAL1 Interacts with SNLs through its own CW Domain and the PAH Domains of SNLs

Both VAL1 and SNLs proteins play essential roles in seed maturation and flowering transition by repressing genes through mediating histone deacetylation, indicating that these proteins may work together. To validate whether VAL1 interacts with SNLs physically, we first tested the interaction between VAL1 and SNLs by pair-wise Y2H assay. The full-length ORF (Open Reading Frame) of *VAL1* was cloned into pGADT7, and the full-length ORFs of *SNLs* (*SNL1*–*SNL6*) were cloned into pGBKT7. The transformed yeast cells were spotted onto a stringent selection medium lacking Trp, Leu, His, and Ade or a non-selective medium lacking Trp and Leu. Yeast cells co-expressing VAL1 and SNLs (SNL1–SNL6) were able to grow on the selective quadruple-dropout medium, indicating their positive interactions (Figure 1A). 

In order to identify the specific regions of VAL1 and SNLs that are involved in the interaction, we analyzed the VAL1 and SNL3 peptide sequences using the INTERPRO database (http://www.edi.ac.uk/interpro, accessed on 15 June 2020). VAL1 contains four different functional domains including a plant homeodomain (PHD, aa 36–98), a B3 domain (B3, aa 293–396), a zinc-finger CW domain (CW, aa 538–588) and an ethylene-responsive element binding factor-associated amphiphilic repression domain (EAR, aa 721–742). SNL3 harbors three types of domains, including the three paired amphipathic helix domains (PAH1, aa 7–77; PAH2, aa 90–163; PAH3, aa 286–351), an HDAC interaction domain (HID, aa 426–526) and a C-terminal domain (C-terminal, aa 1053–1299) (Figure 1B, left). Then, we generated seven fragments of *VAL1*, including the *VAL1-PHD* (aa 1–207), *VAL1-B3* (aa 208–457), *VAL1-CW* (aa 458–673), *VAL1-EAR* (aa 674–790), *VAL1-PHD.B3* (aa 1–457), *VAL1-B3.CW* (aa 208–673) and *VAL1-CW.EAR* (aa 458–790), and five fragments of *SNL3*, including the *SNL3-PAH* (aa 1–390), *SNL3-HID* (aa 391–850), *SNL3-C terminal* (aa 851–1326), *SNL3-ΔC* (aa 1–850) and *SNL3-**ΔN* (aa 391–1326) (Figure 1B, left). Subsequently, the above fragments of *VAL1* and *SNL3* were cloned into pGADT7 and pGBKT7 to perform Y2H analysis. As a result, SNL3 could interact with VAL1-CW, VAL1-B3. CW and VAL1-CW.EAR, but not with VAL1-PHD, VAL1-B3, VAL1-EAR, and VAL1-PHD.B3 (Figure 1B, right and top), suggesting that the CW domain of VAL1 is responsible for the interaction between VAL1 and SNL3. In addition, VAL1 interacted with SNL3-PAH and SNL3-ΔC, but not with SNL3-HID, SNL3-C terminal and SNL3-ΔN (Figure 1B, right and bottom), indicating that the PAH domain of SNL3 is responsible for the interaction between VAL1 and SNL3. Therefore, it could be concluded that the CW domain of VAL1 and the PAH domain of SNL3 are responsible for the interaction between the two proteins. To further confirm this conclusion, we generated the PAH domain-containing constructs of other *SNLs*, including the *SNL1-PAH* (aa 1–450), *SNL2-PAH* (aa 1–430), *SNL4-PAH* (aa 1–420), *SNL5-PAH* (aa 1–250) and *SNL6-PAH* (aa 1–300) for Y2H analysis (Appendix A). As a result, the CW domain of VAL1 interacted with the PAH domains of all SNLs (SNL1–SNL6) (Figure 1C). A pair-wise Y2H assay also validated the interactions between VAL1-CW and intact SNLs (Appendix A), and between intact VAL1 and SNLs-PAH (Appendix A). Taken together, VAL1 can directly interact with SNLs (SNL1–SNL6), and the CW domain of VAL1 and the PAH domains of SNLs are responsible for the interactions between them as indicated by the Y2H assay.

### 2.2. In vitro Pull-down Assay Validated the Interaction between VAL1-CW and SNLs

To further confirm the interaction between VAL1 and SNLs, an in vitro pull-down assay was performed to test their interaction. Firstly, the full-length ORF of *VAL1* and the CW domain of *VAL1* (aa 458–673) were amplified and cloned into the vector pGEX4T-1 containing a GST tag, and the full-length ORFs of *SNLs* were amplified and cloned into the vector pET28a containing a His tag. Then, the GST-VAL1, GST-VAL1-CW and His-SNLs fusion proteins were induced to express in *E. coli* cells under a series of induction conditions. As a result, the GST-VAL1-CW and His-SNLs fusion proteins were successfully induced to express in *E. coli* cells, while the GST-VAL1 was not. Subsequently, the induced His-SNLs proteins were incubated with the GST-VAL1-CW or GST protein, and then the protein samples were immunoprecipitated with anti-GST antibodies and immunoblotted with anti-His antibodies. After Western blot analysis using anti-His antibodies, the expected His-SNLs bands could be detected with the GST-VAL1-CW, but not with the GST control, indicating that the CW domain of VAL1 specifically interacts with intact SNLs (SNL1–SNL6) (Figure 2). 

To determine whether the CW-domain of VAL1 interacts with the PAH domains of SNLs in vitro, the CW domain of *VAL1* (aa 458–673) was cloned into the vector pET28a, and the PAH domains of *SNLs* (*SNL1-PAH*, aa 1–450; *SNL2-PAH*, aa 1–430; *SNL3-PAH*, aa 1–390; *SNL4-PAH*, aa 1–420; *SNL5-PAH*, aa 1–250; *SNL6-PAH*, aa 1–300) were cloned into the vector pGEX4T-1. After successful induction of expression of the GST-SNLs-PAH and His-VAL1-CW fusion proteins, pull-down experiments were performed by immunoprecipitating with anti-GST antibodies and immunoblotting with anti-His antibodies. After Western blot analysis using anti-His antibodies, the expected His-VAL1-CW bands (~25 kDa) could be detected with the GST-SNLs-PAH, but not with the GST control, indicating that the CW domain of VAL1 specifically interacts with the PAH domains of SNLs (SNL1–SNL6) in vitro (Appendix A). Taken together, the CW domain of VAL1 can directly interact with intact SNLs and SNLs-PAH.

### 2.3. LCI Assays Verified That VAL1 Interacts with SNLs through the CW Domain of VAL1 and the PAH Domains of SNLs

Based on the above results of Y2H and pull-down assays, to further confirm whether VAL1 interacts with SNLs through the CW domain of VAL1 and the PAH domains of SNLs in plants, a firefly luciferase complementation imaging (LCI) assay was performed as described [20] to test their interactions in *Nicotiana benthamiana*. Intact *VAL1* and *VAL1-CW* (aa 458–673) were in-frame fused with the N-terminal half of luciferase (*nLUC*) [21], and intact *SNLs* and *SNLs-PAH* (*SNL1-PAH*, aa 1–450; *SNL2-PAH*, aa 1–430; *SNL3-PAH*, aa 1–390; *SNL4-PAH*, aa 1–420; *SNL5-PAH*, aa 1–250; *SNL6-PAH*, aa 1–300) were in-frame fused with the C-terminal half of luciferase (*cLUC*) [21]. An interaction between two proteins brings the two halves of the luciferase together, leading to the firefly luciferase enzymatic activity, which can be detected using a low-light imaging device. As shown in Figure 3, VAL1 could interact with SNLs and VAL1-CW could interact with SNLs-PAH in *N*. *benthamiana*. In addition, intact VAL1 could interact with SNLs-PAH and intact SNLs could interact with VAL1-CW in *N*. *benthamiana* (Appendix A). These results further confirmed that VAL1 interacts with SNLs through the CW domain of VAL1 and the PAH domains of SNLs in plants.

### 2.4. CO-IP Assay Validated the Interaction between VAL1 and SNLs

To further confirm whether VAL1 interacts with SNLs in vivo, we carried out a co-immunoprecipitation assay in *N*. *benthamiana*. The full-length ORF of *VAL1* was cloned into a modified pMDC32-based vector containing a 3× *FLAG* tag after the *CaMV 35S* promoter and the full-length ORFs of *SNLs* were cloned into a modified pMDC32-based vector containing a 3× *HA* tag after the *CaMV 35S* promoter. The VAL1-FLAG fusion protein was then co-expressed with SNLs-HA fusion proteins in *N*. *benthamiana.* After immunoprecipitation using the anti-HA antibody, a Western blot was performed using the anti-FLAG antibody to detect VAL1-FLAG. As shown in Figure 4, the expected VAL1-FLAG bands (~90 kDa) could be detected in the presence of 35S::SNLs-HA, but not in the absence of 35S::SNLs-HA, indicating that VAL1 specifically interacts with SNLs (SNL1–SNL6) in vivo. 

### 2.5. VAL1 and SNLs Are Expressed in Seedlings and Their Encoding Proteins Localize in the Nucleus

To gain insight into the formation of the VAL1-SNL complex in plants, we performed an expression pattern analysis of *VAL1* and *SNLs* using qRT-PCR in *Arabidopsis* seedlings that were 2 weeks old. As shown in Figure 5A, both *VAL1* and *SNLs* were expressed in the aerial parts and roots of *Arabidopsis* seedlings. In addition, we performed a subcellular localization analysis of VAL1 and SNLs with transient expression assays in *Arabidopsis* protoplasts. As shown in Figure 5B, the green fluorescent protein (GFP) signals of the VAL1-GFP and SNLs-GFP fusion proteins were localized to the nucleus and overlapped closely with the mCherry signal of a known nuclear protein, the OsGhd7-mCherry fusion protein. The co-expression of *VAL1* and *SNLs* and co-localization of their encoding proteins support the possibility of interaction between VAL1 and SNLs in plants.

## 3. Discussion

The *VP1/ABI3-LIKEs* (*VALs*) family comprises three members designated as *VAL1*, *VAL2* and *VAL3* [7]. *VAL3* lacks an intact PHD domain and has lower expression levels than *VAL1* and *VAL2* in *Arabidopsis* seedlings, and the *val1 val3* and *val2 val3* double mutants do not exhibit discernible phenotypes like *val1* and *val2* single mutants; however, the double mutant *val1 val2* exhibits serious developmental defects in seedlings [2,7]. Currently, most studies on *VALs* have been focused on *VAL1* and *VAL2*, which regulate plant development in a functionally redundant manner [2,3,7,8,11,12]. In this study, our Y2H assay demonstrated that VAL1 interacts with SNLs through the CW domain of VAL1 and the PAH domains of SNLs. We also tested the interaction of pairwise combinations of VAL2/SNL1, VAL2/SNL1-PAH, VAL2-CW/SNL1, VAL2-CW/SNL1-PAH, VAL2/SNL3, VAL2/SNL3-PAH, VAL2-CW/SNL3, and VAL2-CW/SNL3-PAH in Y2H assay, and no interaction in the above pairwise combinations was observed (Appendix A). Amino acid sequence alignment of VAL1-CW (aa 458–673) and VAL2-CW (aa 437–655) shows that the CW domains of VAL1 and VAL2 are composed of approximately 50 amino acid residues with four conserved cysteine (C) and two conserved tryptophan (W) residues (Appendix A); however, the amino acid identity between VAL1-CW and VAL2-CW is approximately 55% (Appendix A), and the different amino acid residues may contribute to different structures of VAL1-CW and VAL2-CW, which causes the differences in interaction of VAL1 and VAL2 with SNLs. In a previous study, the Y2H assay also revealed that HDA19 does not interact with VAL1 but interacts with VAL2 through its CW domain [13]. These results suggest that the CW domains of VAL1 and VAL2 may interact with different proteins though they are homologous to each other. Moreover, VAL1 can interact with itself and VAL2, suggesting that VAL1 and VAL2 can form homodimers and heterodimers [3], and HDA19 interacts with SNL1 in vitro and in planta [13], these results indicate that the VALs (VAL1/2), the SNLs (SNL1–SNL6) and HDA19 may interact with each other to form a complex and work together in some pathways.

The SIN/HDAC complex generally consists of several core proteins including SWI-INDEPENDENT3 (SIN3), histone deacetylase (HDAC), the SIN3-associated proteins (SAP18 and SAP30) and the retinoblastoma (Rb)-associated proteins (RbAp46 and RbAp48) [15]. In *Arabidopsis*, there are six SIN3 homologs known as SNL1–SNL6 [16], two HDAC1 homologs known as HDA6 and HDA19 [22], and five RbAp46/48 homologs known as MSI1-MSI5 (MSI for MULTICOPY SUPPRESSOR OF *IRA1*) [23]. Several recent studies have revealed that VAL proteins interact with the components of the SIN/HDAC complex. For instance, Y2H and Co-IP assays showed that HDA19 interacts with HSL1 (also named VAL2) [13] and affinity purification of HA-VAL1 from *Arabidopsis* seedlings revealed that VAL1 forms a complex with AtSAP18 [2]. In this study, we provided substantive data validating that VAL1 interacts with SNLs. Since SNLs function as scaffold proteins in the SIN/HDAC complex, we speculate that SNLs may act as a bridge protein to connect the VAL1 and SIN/HDAC complex. VAL proteins can specifically bind the RY motif through their B3 domains in *Arabidopsis* and in rice [2,3,24,25]. In addition, numerous studies have revealed that the VAL (VAL1/2) proteins and the components of the SIN/HDAC complex, including SNLs, MSIs, HDA6, HDA19 and AtSAP18 proteins, are all involved in repression gene expression through mediating histone deacetylation in seed maturation and flowering transition [2,3,6,7,8,9,11,12,13,23,26,27,28,29,30,31,32,33,34]. We speculate that the SIN/HDAC complex may be recruited to the target genes containing RY motifs by VAL1. The important roles of the VALs and SNLs family in seed maturation and flowering transition of *Arabidopsis* need to be elucidated by genetic analysis with the *vals, snls* and *vals snls* mutants, chromatin immunoprecipitation (CHIP) assay and histone modification analysis of the candidate genes.

## 4. Methods

### 4.1. Plasmid Construction

All plasmids used in this study were assembled using the Gibson assembly method [35]. In brief, primers were designed to contain 15–21 nucleotides overlapping regions with adjacent DNA fragments of the target vectors. The PCR products amplified by primer pairs and the target vectors digested by appropriate restriction endonucleases were purified and quantified using NanoDrop 2000 (Thermo Fisher Scientific, Waltham, MA, USA). Fragment and vector were mixed in a molar ratio of 3:1. Then, 0.5 U of T5 endonuclease (New England Biolabs (Beijing) LTD., Beijing, China), 0.5 μL of buffer4 (New England Biolabs (Beijing) LTD., Beijing, China), and ddH_2_O were added into the mixture to a final volume of 5 μL. All reagents were mixed and incubated on ice for 5 min, and then added to chemically competent *E. coli* cells. After incubation on ice for 30 min, heat-shocking for 1 min at 42 ℃, and holding on ice for 5 min, 200 μL of LB medium was added to the mixture above and recovered for 1 h at 37 °C with shaking (220 rpm). Cells were plated on LB agar plates containing kanamycin or ampicillin, and recombinants were selected by colony PCR and confirmed using Sanger sequencing (TsingKe Biological Technology, Wuhan, China). Primers used for plasmid construction in this study are listed in Appendix A.

### 4.2. Yeast Two-hybrid Assays

Yeast two-hybrid assays were performed according to the manufacturer’s instructions for the Matchmaker GAL4-based two-hybrid system 3 (Clontech). The full-length and different domain-containing constructs of *VAL1* cDNA were subcloned into the pGADT7 vector, whereas the full-length and different domain-containing constructs of *SNLs* cDNA were subcloned into the pGBKT7 vector. All paired constructs were co-transformed into yeast strain *AH109* using the lithium acetate method, and the transformed yeast cells were spotted onto a stringent selection medium lacking Trp, Leu, His, and Ade, or a non-selective medium lacking Trp and Leu, to test the possible interactions. Empty vectors were used as negative controls. The transformed yeast cells were grown for 3–5 d at 30 °C before representative images were taken.

### 4.3. Pull-Down Assays

To generate GST and His tag fusion proteins, the *VAL1-CW* and the *SNLs-PAH* were amplified and cloned into the vector pGEX4T-1, and the full-length ORFs of *SNLs* and the *VAL1-CW* were amplified and cloned into the vector pET28a. Then, the GST-VAL1-CW, GST-SNLs-PAH, His-SNLs, His-VAL1-CW, and GST proteins were expressed in cells of *E. coli* strain BL21 (DE3) (Transetta) under induction with 0.5 mM isopropyl-beta-D-thiogalactoside and shaken at 16 °C for 14 h. Proteins were collected and resuspended in PBS buffer (137 mM NaCl, 2.7 mM KCl, 10 mM Na_2_HPO_4_, 2 mM KH_2_PO_4_, pH 7.5, and 1× protease inhibitor cocktail). The resuspended liquid was sonicated to break the cells until the liquid was clear (cycles of 4 s on and 2 s off) and then centrifuged at 4 °C and 12,000 rpm for 30 min to collect the supernatant for pull-down analysis. To detect the protein–protein interaction with an in vitro pull-down assay, roughly equal amounts of GST or GST-VAL1-CW proteins were mixed with His-SNLs, and GST or GST-SNLs-PAH proteins were mixed with His-VAL1-CW. The mixed supernatants were incubated with glutathione beads (SA008010, Smart-Lifesciences, Changzhou, China) or Ni Singarose 6FF (AGM90046, AOGMA, Shanghai, China) overnight at 4 °C. The beads were washed five times with PBS buffer (137 mM NaCl, 2.7 mM KCl, 10 mM Na_2_HPO_4_, 2 mM KH_2_PO_4_, pH 7.5) and boiled with 200 μL of protein loading buffer (50 mM Tris-HCl, 2% SDS, 10% glycerol, 0.1% bromophenol blue, and 1% β-mercaptoethanol) at 100 °C for 10 min. The proteins were separated in 10% SDS-PAGE gels and detected by Western blot analysis using anti-GST antibody (PM013, MBL BEIJING BIOTECH CO., LTD., Beijing, China, 1:4000) or anti-His antibody (M136-3, MBL BEIJING BIOTECH CO., LTD., Beijing, China, 1:4000).

### 4.4. LCI Assays

The detection of protein–protein interactions by the LCI assays was performed as previously described [20]. The full-length and CW domain-containing fragments of *VAL1* cDNA were in-frame fused with the N-terminal half of luciferase (*nLUC*) [21], whereas the full-length and PAH domain-containing fragments of *SNLs* cDNA were in-frame fused with the C-terminal half of luciferase (*cLUC*) [21]. The resulting binary expression vectors were transformed into *Agrobacterium* strain GV3101 (containing the pSoup-p19 plasmid). *Agrobacterium* cells carrying various expression vectors were co-infiltrated into tobacco (*Nicotiana benthamiana*) leaves with appropriate controls. After the infiltration, plants were placed at 22 °C for 48 h and the fluorescent images were collected with a low-light cooled charge-coupled device (CCD) imaging apparatus (Tanon 5200, Beijing, China).

### 4.5. CO-IP Assays

To generate *35S::VAL1-FLAG* and *35S::SNLs-HA* constructs, the full-length cDNA of *VAL1* and *SNLs* was subcloned into a modified pMDC32-based vector which was inserted with 3x *FLAG* or 3x *HA* after the *CaMV 35S* promoter. Subsequently, the constructs were introduced into *Agrobacterium* strain *GV3101* and infiltrated into tobacco (*Nicotiana benthamiana*) as described previously [36]. CO-IP assay was performed as previously reported [36]. Briefly, two days after infiltration, tobacco leaves were harvested and ground in liquid nitrogen. Proteins were extracted in an extraction buffer (50 mM Tris-HCl, pH 7.5, 150 mM NaCl, 1 mM EDTA, 1 mM DTT, 10% glycerol, 1mM PMSF, and 0.1% Triton X-100). Cell debris was pelleted using centrifugation at 13,000× *g* for 12 min. The supernatant was incubated with 15 μL of anti-HA magnetic beads (L-1009, BIOLINKEDIN, Shanghai, China) at 4 °C overnight. Then, the beads were centrifuged and washed four times with a washing buffer (20mM Tris-HCl pH 8.0, 150mM NaCl), and boiled with 30 μL of protein loading buffer (50 mM Tris-HCl, 2% SDS, 10% glycerol, 0.1% bromophenol blue, and 1% β-mercaptoethanol) at 100 °C for 10 min. The proteins were separated in 10% SDS-PAGE gels and detected with Western blot analysis by immunoblotting using anti-Flag antibody (M185-3S, MBL BEIJING BIOTECH CO., LTD, Beijing, China, 1:4000) and anti-HA antibody (M132-3, MBL BEIJING BIOTECH CO., LTD, Beijing, China, 1:4000).

### 4.6. qRT-PCR Assays

*Arabidopsis* ecotype Columbia (Col) was grown in a climate incubator at 22 °C under long-day conditions (16 h light/8 h dark) for 2 weeks, then the aerial parts and roots of *Arabidopsis* seedlings were sampled for RNA extraction. After reverse-transcription, qRT-PCR was carried out to detect the gene relative expression levels, with the *Tub2* gene (At5g62690) as an internal reference. Each sample had three biological replicates, and there were three technical replicates for each biological replicate. All the primers used for qRT-PCR in this study are listed in Appendix A.

### 4.7. Subcellular Localization Assays

Transient expression assays were performed in *Arabidopsis* protoplasts to determine the subcellular localization of VAL1 and SNLs. The full-length cDNA of *VAL1* and *SNLs* was subcloned into the transient expression vector pAN580 to produce the p35S::VAL1-GFP or p35S::SNLs-GFP fusion constructs. *Arabidopsis* protoplast preparation and the transient expression assays were performed as previously described [37]. Fluorescence signals were observed using a Zeiss LSM880 laser scanning confocal microscope.

### 4.8. Accession Numbers

Sequence data from this article can be found in the *Arabidopsis* Genome Initiative or GenBank/EMBL databases under the following accession numbers: *VAL1* (At2g30470), *SNL1* (At3g01320), *SNL2* (At5g15020), *SNL3* (At1g24190), *SNL4* (At1g70060), *SNL5* (At1g59890), and *SNL6* (At1g10450).

## 5. Conclusions

In *Arabidopsis,* the transcription repressor VAL1 can recognize and bind to the RY motif via its plant-specific B3 domain and repress gene expression through epigenetic mechanisms. The *Arabidopsis* transcription coregulator SNLs are scaffold proteins for the assembly of the HDAC complex, which is associated with gene repression. In this study, we provided substantive data to show that VAL1 interacts with SNLs through its own CW domain and the PAH domains of SNLs. These protein–protein interactions suggest that the SNLs-containing HDAC complex may be recruited to the target genes with the RY motif by VAL1, providing new insights into the regulatory mechanisms of seed maturation and flowering transition mediated by VAL1 and SNLs.

## Figures and Tables

**Figure 1 ijms-23-06987-f001:**
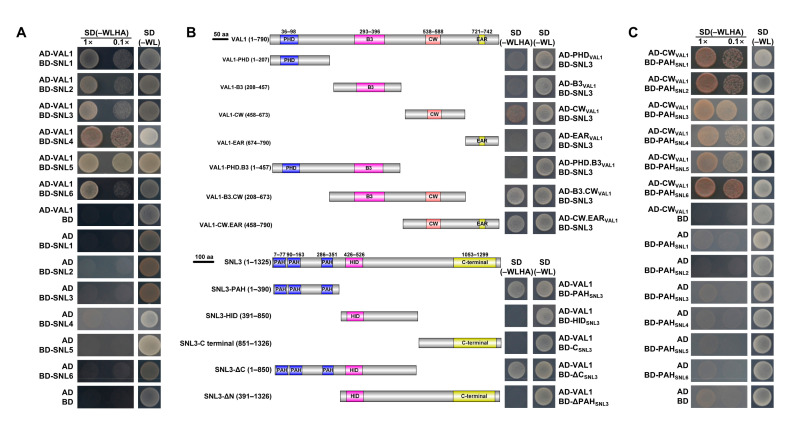
VAL1 interacts with SNLs through the CW domain of VAL1 and the PAH domains of SNLs in yeast cells. (**A**) Intact VAL1 interacts with complete SNLs (SNL1–SNL6) in yeast cells. Full-length of *VAL1* and *SNLs* were fused to the GAL4 activating domain (AD) and binding domain (BD), respectively. The transformed yeast cells were spotted onto a stringent selection medium lacking Trp, Leu, His, and Ade (–WLHA) or a non-selective medium lacking Trp and Leu (–WL; control). (**B**) Schematic presentation of the truncated regions of VAL1 and SNL3, and the CW domain of VAL1 (aa 458–673) and the PAH domain of SNL3 (aa 1–390) were responsible for their interaction. Shown below are the conserved PHD, B3, CW, and EAR domains of VAL1; the PAH and HID domains of SNL3; The VAL1-PHD.B3, VAL1-B3.CW, and VAL1-CW.EAR representing truncated CW and EAR domain, truncated PHD and EAR domain, truncated PHD and B3 domain regions of VAL1, respectively; the SNL3-C, SNL3-ΔC, and SNL3-ΔN representing the C-terminal, truncated C-terminal, and truncated N-terminal regions of SNL3, respectively; and the interactions of each truncated fragment of VAL1 and SNL3 with each other’s full-length in yeast cells. The full-length or regions containing different domains of VAL1 and SNL3 were fused to the GAL4 activating domain (AD) and binding domain (BD), respectively. The transformed yeast cells were spotted onto a stringent selection medium lacking Trp, Leu, His, and Ade (–WLHA) or a non-selective medium lacking Trp and Leu (–WL; control). (**C**) The CW domain of VAL1 interacts with the PAH domains of SNLs in yeast cells. The CW domain of VAL1 and the PAH domains of SNLs were fused to the GAL4 activating domain (AD) and binding domain (BD), respectively. The transformed yeast cells were spotted onto a stringent selection medium lacking Trp, Leu, His, and Ade (–WLHA) or a non-selective medium lacking Trp and Leu (–WL; control).

**Figure 2 ijms-23-06987-f002:**
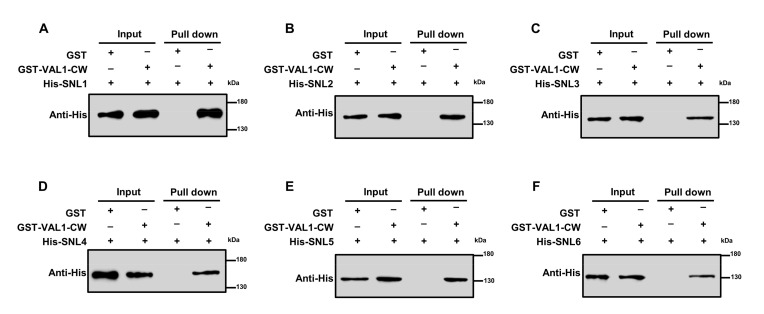
In vitro pull-down assay validated that the CW domain of VAL1 interacts with SNLs. (**A**–**F**) Induced GST-VAL1-CW (aa 458–673) or GST protein was incubated with His-SNLs (SNL1-SNL6). All protein samples were immunoprecipitated with anti-GST antibodies and immunoblotted with anti-His antibodies. The symbols ‘‘−’’and ‘‘+’’ represent the absence and presence of corresponding proteins.

**Figure 3 ijms-23-06987-f003:**
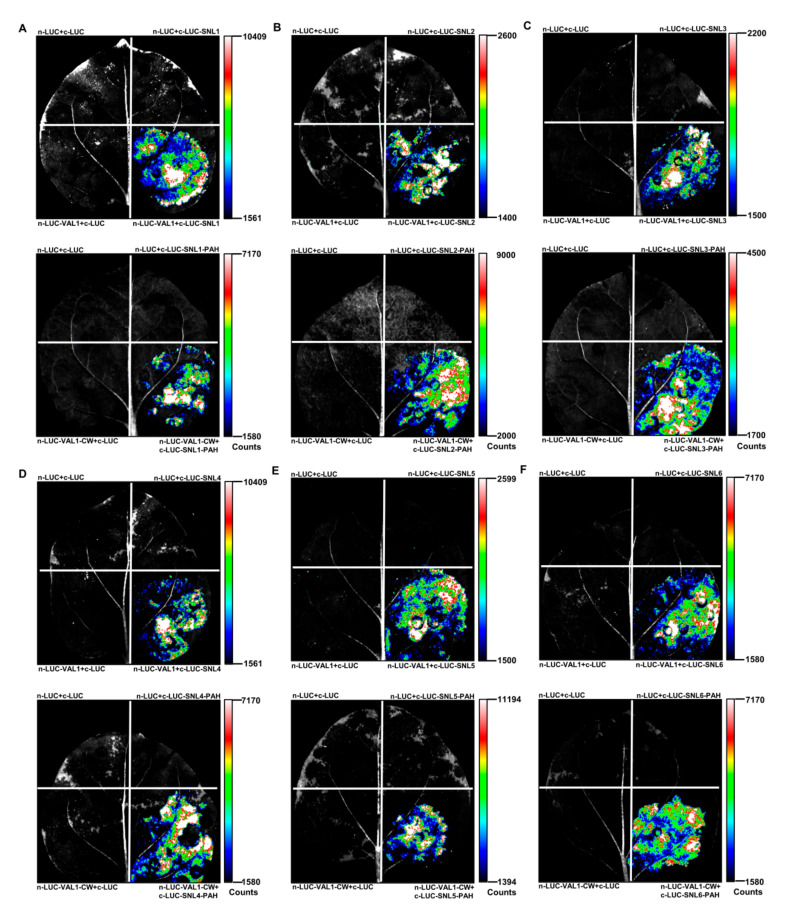
LCI assay verified that VAL1 interacts with SNLs through the CW domain of VAL1 and the PAH domains of SNLs in tobacco. (**A**–**F**) VAL1 interacts with SNLs (SNL1–SNL6), and its CW domain (aa 458–673) interacts with the PAH domains of SNL1 (aa 1–450), SNL2 (aa 1–430), SNL3 (aa 1–390), SNL4 (aa 1–420), SNL5 (aa 1–250) and SNL6 (aa 1–300) in the leaf epidermal cells of *N. benthamiana*. The intact and CW domain of *VAL1* were fused to the *n-LUC* fragment, and the intact and PAH domains of *SNLs* (*SNL1*–*SNL6*) were fused to the *c-LUC* fragment.

**Figure 4 ijms-23-06987-f004:**
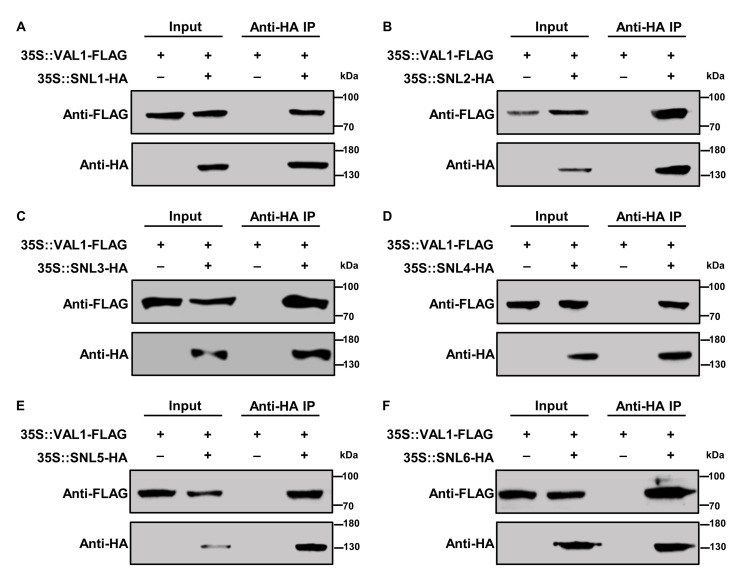
In vivo Co-IP assay validated the interaction between VAL1 and SNLs. (**A**–**F**) Protein extracts obtained from tobacco leaves infiltrated with *Agrobacterium* suspensions harboring 35S::VAL1-FLAG and 35S::SNLs-HA constructs. All protein samples were immunoprecipitated (IP) using an anti-HA antibody and then immunoblotted with anti-FLAG. The symbols ‘‘−’’and ‘‘+’’ represent the absence and presence of the corresponding proteins.

**Figure 5 ijms-23-06987-f005:**
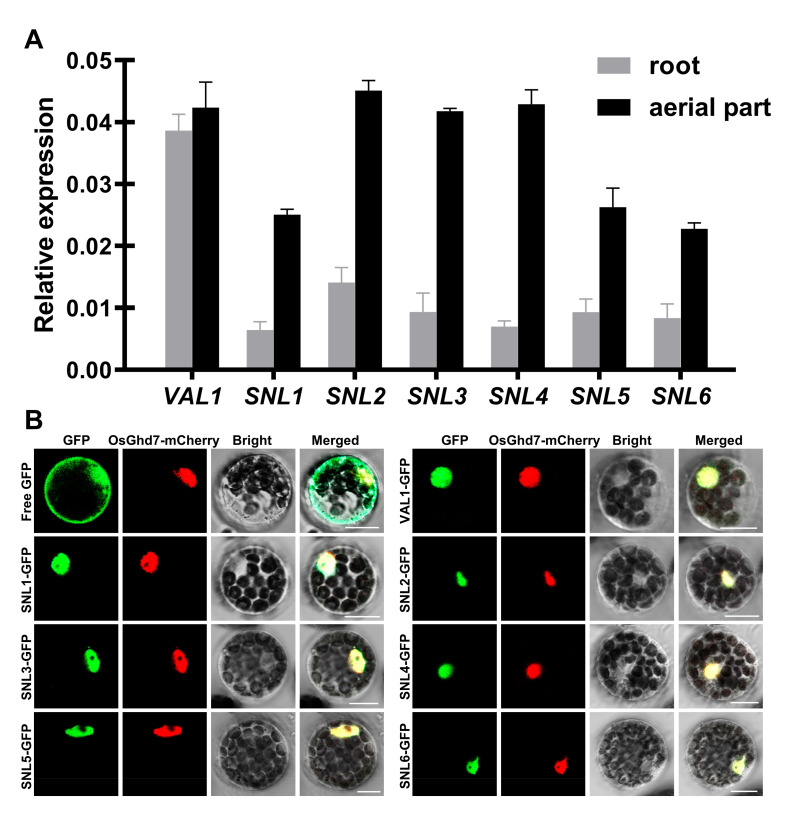
Expression pattern analysis of *VAL1* and *SNLs*, and subcellular localization of the VAL1 and SNLs proteins. (**A**) Expression pattern analysis of *VAL1* and *SNLs* in *Arabidopsis* seedlings that were 2 weeks old. Transcript abundance was determined using qRT-PCR with *Tub2* as a reference. Values are presented as means ± SD (*n* = 3). (**B**) Subcellular localization of the VAL-GFP and SNLs-GFP fusion proteins in *Arabidopsis* protoplasts. OsGhd7-mCherry was used as a nuclear marker. Free GFP was used as the control. VAL1-GFP or SNLs-GFP was co-transformed into *Arabidopsis* protoplasts with OsGhd7-mCherry, and overlapping GFP and mCherry signals were observed in the nucleus. Bars = 10 µm.

## Data Availability

Not applicable.

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
