# Peer review of "Interaction Analysis between the Arabidopsis Transcription Repressor VAL1 and Transcription Coregulators SIN3-LIKEs (SNLs)"

_ijms, 2022, doi:10.3390/ijms23136987_

Round 1

Reviewer 1 Report

In the present manuscript "Interaction analysis between the Arabidopsis transcription repressor VAL1 and transcription co-regulators SIN3-LIKEs 3 (SNLs)", Chen et al investigate the interaction between the DNA-binding B3 domain proteins VAL1 and the six homologs of the SWI-INDEPENDENT3 (SIN3)-LIKE (SNLs). SNLs are scaffold proteins in the SIN/HDAC complex. 

The authors confirm the interaction between the different proteins by four complementary ways: Y2H, in which they show that the interaction involves the CW domain of VAL1 and the PAH domain of the SNL homologues.

They then confirm the interaction by in vitro pull-down showing interaction between the CW domain of VAL1 and full-length SNL proteins. They further show interaction in planta (N. benthamiana) by firefly luciferase complementation imaging and finally co-immunoprecipitation of proteins expressed in N. benthamiana.

The study is well conducted and brings solid evidence for a connection between VAL1 and the SIN / HDAC complex. As the authors have pointed out in the discussion, further genetic evidence on the biological role of this interaction would have been interesting to include. 

I found the observation that VAL2 does not interact with SNLs, mentioned in the discussion, quite interesting and it would certainly be useful to include this negative dataset as well, in particular in link with the known interaction between HDA19 and VAL2.

A sequence alignment of the CW domains of VAL1 and VAL2 would also be interesting to include to discuss more in depth the reason for the differences in interaction of VAL1 and VAL2 with SNLs. 

Minor points: Please check carefully for some typos: e.g. line 218 (35::SNLs should the 35S), line 33 epigentic, line 

Author Response

Reviewer #1: In the present manuscript "Interaction analysis between the Arabidopsis transcription repressor VAL1 and transcription co-regulators SIN3-LIKEs 3 (SNLs)", Chen et al investigate the interaction between the DNA-binding B3 domain proteins VAL1 and the six homologs of the SWI-INDEPENDENT3 (SIN3)-LIKE (SNLs). SNLs are scaffold proteins in the SIN/HDAC complex. 

The authors confirm the interaction between the different proteins by four complementary ways: Y2H, in which they show that the interaction involves the CW domain of VAL1 and the PAH domain of the SNL homologues.

They then confirm the interaction by in vitro pull-down showing interaction between the CW domain of VAL1 and full-length SNL proteins. They further show interaction in planta (N. benthamiana) by firefly luciferase complementation imaging and finally co-immunoprecipitation of proteins expressed in N. benthamiana.

The study is well conducted and brings solid evidence for a connection between VAL1 and the SIN / HDAC complex. As the authors have pointed out in the discussion, further genetic evidence on the biological role of this interaction would have been interesting to include. 

Response 1: We sincerely thank the reviewer for these encouraging and positive comments

I found the observation that VAL2 does not interact with SNLs, mentioned in the discussion, quite interesting and it would certainly be useful to include this negative dataset as well, in particular in link with the known interaction between HDA19 and VAL2.

A sequence alignment of the CW domains of VAL1 and VAL2 would also be interesting to include to discuss more in depth the reason for the differences in interaction of VAL1 and VAL2 with SNLs. 

Response 2: Thanks for these suggestions. We have added the Figure S4 to presented the negative interaction dataset between VAL2 and SNL1/3 and the amino sequence alignment of the CW domains of VAL1 and VAL2. We also add more discussion on the reason for the differences interaction of VAL1 and VAL2 with SNLs in the revised discussion part.

Minor points: Please check carefully for some typos: e.g. line 218 (35::SNLs should the 35S), line 33 epigentic, line 

Response 3: Thanks for your reminding. We have checked the text carefully again and found some typos and revised them in the revised text.

Reviewer 2 Report

This work is completed and high-quality. The methods used are the appropriate and they explain the conclusions. 

Author Response

Reviewer #2: This work is completed and high-quality. The methods used are the appropriate and they explain the conclusions. 

Response: We sincerely thank the reviewer for these positive comments

Reviewer 3 Report

The interactions between VAL1 and SNLs are well demonstrated by different approaches, both in vitro and in vivo (on transient expression in N benthamiana); however,  the authors could include something about the physiological relevance of these interactions, such as in which tissues they are located and which interaction is more relevant between the different SNLs, or inquire more about the formation and composition of the VAL1-SNL complex

Author Response

Reviewer #3: The interactions between VAL1 and SNLs are well demonstrated by different approaches, both in vitro and in vivo (on transient expression in N benthamiana); however,  the authors could include something about the physiological relevance of these interactions, such as in which tissues they are located and which interaction is more relevant between the different SNLs, or inquire more about the formation and composition of the VAL1-SNL complex.

Response: Thanks you for these comments. As suggestions we performed gene expression pattern analysis and protein subcellular localization of VAL1 and SNLs, and the data was shown in Figure 5. The co-expression in seedling and co-localization in the nucleus of VAL1 and SNLs provide the physiological relevance of their interaction.

Round 2

Reviewer 3 Report

The authors should discute the gene expresion differences between aerial and root tissues, as well as the differences between VAL1 and SNLs in roots. Are these differences relevant for plant physiology? What role could VAL-SNLs interaction have in aerial and root tissues?